# Natural Compounds and Derivates: Alternative Treatments to Reduce Post-Harvest Losses in Fruits

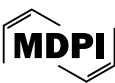

Edson Rayón-Díaz [1], Luis G. Hernández-Montiel [2], Jorge A. Sánchez-Burgos [1], Victor M. Zamora-Gasga [1], Ramsés Ramón González-Estrada [1] and Porfirio Gutiérrez-Martínez [1,*]

[1] Laboratorio Integral de Investigaciones en Alimentos, Tecnológico Nacional de Mexico/Instituto Tecnológico de Tepic, Avenida Tecnológico #2595, Col. Lagos del Country, Tepic 63175, Nayarit, Mexico; edrayondi@ittepic.edu.mx (E.R.-D.); jsanchezb@ittepic.edu.mx (J.A.S.-B.); vzamora@tepic.tecnm.mx (V.M.Z.-G.); ramgonzalez@ittepic.edu.mx (R.R.G.-E.)

[2] Nanotechnology & Microbial Biocontrol Group, Centro de Investigaciones Biológias del Noroeste, La Paz 23096, Baja California Sur, Mexico; lhernandez@cibnor.mx

* Correspondence: pgutierrez@ittepic.edu.mx

**Abstract:** The effects of phytopathogenic fungi on fruits and vegetables are a significant global concern, impacting various sectors including social, economic, environmental, and consumer health. This issue results in diminished product quality, affecting a high percentage of globally important fruits. Over the last 20 years, the use of chemical products in the agri-food sector has increased by 30%, leading to environmental problems such as harm to main pollinators, high levels of chemical residue levels, development of resistance in various phytopathogens, and health issues. As a response, various organizations worldwide have proposed programs aimed at reducing the concentration of active compounds in these products. Priority is given to alternative treatments that can mitigate environmental impact, control phytopathogens, and ensure low residuality and toxicity in fruits and vegetables. This review article presents the mechanisms of action of three alternative treatments: chitosan, citral, and hexanal. These treatments have the potential to affect the development of various pathogenic fungi found in tropical and subtropical fruits. It is important to note that further studies to verify the effects of these treatments, particularly when used in combination, are needed. Integrating the mechanisms of action of each treatment and exploring the possibility of generating a broad-spectrum effect on the development of pathogenic microorganisms in fruits is essential for a comprehensive understanding and effective management.

**Keywords:** post harvest; essential oils; chitosan; antifungal activity; GRAS substances; bibliometric analyses

## 1. Introduction

Global fruit production during the last five years has generated around 1,882 million tons of produce, according to data from the Food and Agriculture Organization of the United Nations (FAO) up until 2020 [1]. The last recorded global fruit production worldwide amounted to 909.6 million tons [2], contributing to a market size which is valued at USD 551, 100 billion, as reported in 2021 [3].

Management during the post-harvest stage has two main objectives. The first is to maintain the physical integrity and quality of the products. The second focuses on preserving the products for long periods, preventing their nutritional and commercial value from decreasing [4].

In this sense, the United Nations General Assembly established the 2030 Agenda in 2015, which proposed 17 objectives involving the economic, social, and environmental sectors [5]. According to the proposed objectives, it is important to decrease food losses and waste. Additionally, there should be a reduction in the application of chemical products

in the field as well as efforts to mitigate climate change and promote the consumption of residue-free foods [5–7].

However, in the agri-food sector, post-harvest losses range between 40 and 50% of the total production. These losses occur due to limitations in fruit handling methods during classification, packing, storage, transportation, and marketing. Therefore, one of the main agents of damage is the presence of microorganisms, which significantly impact a product's quality [8,9]. Additionally, the inappropriate use of pesticides has increased by 30% in the last 20 years [10].

Naturally, infections by phytopathogenic fungi cause problems in one or more types of plants and fruits [11,12]. The mechanisms of fungal infection are generated by the secretion of enzymes that aid in the adhesion to the surface of the fruit, causing deterioration in its physical characteristics and commercial quality [13]—hence the importance of understanding the survival modes, which are influenced by factors such as temperature conditions, humidity, and, in most cases, the presence of water in the environment [14].

The presence of these pathogens decreases crop yield and quality and causes huge losses in agricultural production [12,15]. In this sense, changes in the chemical composition and nutritional attributes of fruits, caused by fungi infection, induce the generation of acidity, the breakdown of sugar molecules, and increased activity of microbial metabolites [16,17]. In addition, the prevalence rate of these pathogens in food is estimated to be between 60 and 80%. These pathogens primarily belong to the genera Aspergillus, Alternaria, Fusarium, and Penicillium [18].

The control of phytopathogenic fungi through the use of chemical treatments appears to have fungistatic and fungicidal effects; however, in some research works, poor application practices of these products induce resistance mechanisms, residual chemical particles in fruits, and effects on consumer health [19–21].

Therefore, the adoption of novel technologies, such as "Generally Recognized as Safe" (GRAS) substances, has taken an innovative turn. Presently, there are non-polluting integrated control strategies for post-harvest diseases, primarily relying on natural compounds like essential oils and organic compounds such as aldehydes. Additionally, certain biopolymers with harmless and non-toxic attributes, such as chitosan, offer a viable alternative for the food and agricultural industry due to their low residuality. They also contribute to enhancing the defense mechanisms against pathogens during the post-harvest stage [22,23].

In this review, we analyze the impact of alternative treatments, such as chitosan and essential oil extracts, main citral, and hexanal, on post-harvest applications using a bibliometric analysis. We discuss their main properties, effects of their application on fruits, and their role in the agri-food sector and emerging technologies.

## 2. Bibliometric Analysis of Alternative Post-Harvest Treatments

In recent years, growing scientific research has taken various directions of interest; for that reason, scientometrics is a tool based on quantitative methods and analyses based on scientific production [24]. The contribution of bibliometrics is to provide an analysis of objectives and new evaluations with greater rigor, as well as an improvement in the scientific activities of universities and research centers, which go hand in hand with the improvement in technological processes [25,26].

This section contains specifics on the recommended methodology used for this investigation. The primary steps of the proposed method are data search, collection, integration, and cleansing.

### 2.1. Bibliometric Methodology in Data Collection

In this paper, we have used Science Direct data collection to quantify the bibliographic material in related studies and set up the following search profile: Terms = "alternative treatments" AND "postharvest" AND "fruits" AND "pathogens". The search was restricted to materials released between 2010 and 2023 and was carried out in late March 2024. The results of the search were downloaded and saved as an RIS file.

Once the documents had been obtained, we set up the bibliometric analysis, selected seven as the minimal number of occurrences of a keyword, and visualized co-occurrences between keywords with the VOS-viewer software®1.6.19 version.

## 2.2. Findings from the Bibliometric Study of Contemporary Alternative Post-Harvest Treatments

According to a search of Science Direct data, we obtained a total of 3,133 results. We examined the different types of documents, which are shown in Table 1. The statistics shown indicate that 52.89% (*n* = 1657) of the documents were concentrated in one category and had been published as research articles. Book chapters account for 26.27% of the document types, review articles account for 15.29%, encyclopedias account for 3.29%, and other document types, including editorials, correspondence, short communication, etc., account for 2.27%.

**Table 1.** Publications found by document type from 2010 to 2023.

| Type of Publication | Total of Publications | Percentage of Total Publication (%) |
|---|---|---|
| Research articles | 1657 | 52.89% |
| Book chapters | 823 | 26.27% |
| Review articles | 479 | 15.29% |
| Encyclopedias | 103 | 3.29% |
| Others | 71 | 2.27% |
| Total | 3133 | 100% |

Similarly, the subject area that received the most attention is "Agriculture and Biological Sciences" (n = 2603), indicating a greater propensity for topics related to agriculture as well as knowledge of interactions and effects of treatment–fruit–pathogen in the field of "Biochemistry, Genetics and Molecular Biology" (n = 581). "Chemical Engineering" (n = 393), "Immunology and Microbiology" (n = 313), "Environmental Sciences" (n = 113), and "Material Sciences" (n = 99) were the subjects that carried on the trend of diverse research and sparked more interest.

To improve this study´s interpretation of co-citation analysis (in the past) or bibliographic coupling (in the present), as well as forecast the field´s future development, keyword co-occurrence analysis may be utilized to forecast future research in the area [27]. Less-relevant terms were also manually eliminated to produce more accurate results, and a threshold level of seven keyword occurrences was established. As seen in Figure 1, the bibliometric map has been divided into eight clusters, using the co-occurrence frequency of 342 keywords out of a total of 7315 retrieved terms, with 4256 links, and a link strength of 6048 (indicating the number of cited references which have two elements in common). Each cluster is represented by a different color; the largest circles represent topics with the greatest co-occurrence in the various articles, while, the smallest circles, besides the most distant topics, also represent those with the least co-occurrence in searches related to the aforementioned criteria. Additionally, this presents an overlay visualization covering research works spanning 13 years (2010–2023), as shown in Figure 2, with the color scale indicating the presence of research related to each year, with lighter colors being the potential fields of research.

In addition, an analysis was conducted to explore the relationship between two post-harvest technologies, chitosan and essential oils. This is shown in Figure 3, detailing the co-occurrence results over the period of research spanning from 2010 to 2023, with a total of 2497 results for the analysis of "chitosan", "post-harvest", and "treatments", classified into six clusters (see Figure 3A). Notably, cluster one (red) exhibits the highest co-occurrence, with terms such as "induced resistance" related to words like "gene expression", "oligochitosan", "methyl jasmonate", "defense response", "reactive oxygen species", and "transcriptome". This suggests that the use of chitosan technology may interfere with the natural defense mechanisms of fruits and their interaction with various pathogens. This finding is linked to cluster three (blue), focusing on "fruit quality", and cluster five (purple),

emphasizing "shelf life". Consequently, from the properties and mechanisms of chitosan, it can be inferred that it contributes to extending the shelf life of fruits during the post-harvest period. Furthermore, cluster two (green), highlights the relationship with terms such as "encapsulation", "nanotechnology", "nanoparticles", "antifungal activity", and "biological control", indicating the adaptation of new encapsulation technologies and their combination with treatments derived from essential oils. This relationship also extends to cluster four (light green), emphasizing "edible coatings", and cluster six (light blue), focusing on "antimicrobial" properties.

On the other hand, the essential oils were evaluated under the same analysis, yielding 2798 results, classified into nine clusters (see Figure 3B). Cluster two (green) exhibits the highest co-occurrence, primarily with the term "antifungal activity", which further leads to related terms such as "post-harvest decay", "mycotoxins", "antioxidant enzymes", and "fungi". Consequently, research is this area focuses on studying the efficacy of essential oil treatments against various phytopathogens affecting the post-harvest conditions of fruits and vegetables. From the keyword "post-harvest" (cluster four, light green) derived terms such as "quality", "antimicrobial", "storage", "citral", "eugenol", and "fungicide", indicating investigations focusing on the utilization of specific essential oil compounds during the post-harvest period and storage of fruits due to their antimicrobial activity. Lastly, cluster six (light blue), originating from terms like "encapsulation", "nanotechnology", "post-harvest quality", "agriculture", and "biopolymers", highlights the emphasis on the use of chitosan treatments and essential oils and their integration with nanotechnological tools in modern agriculture.

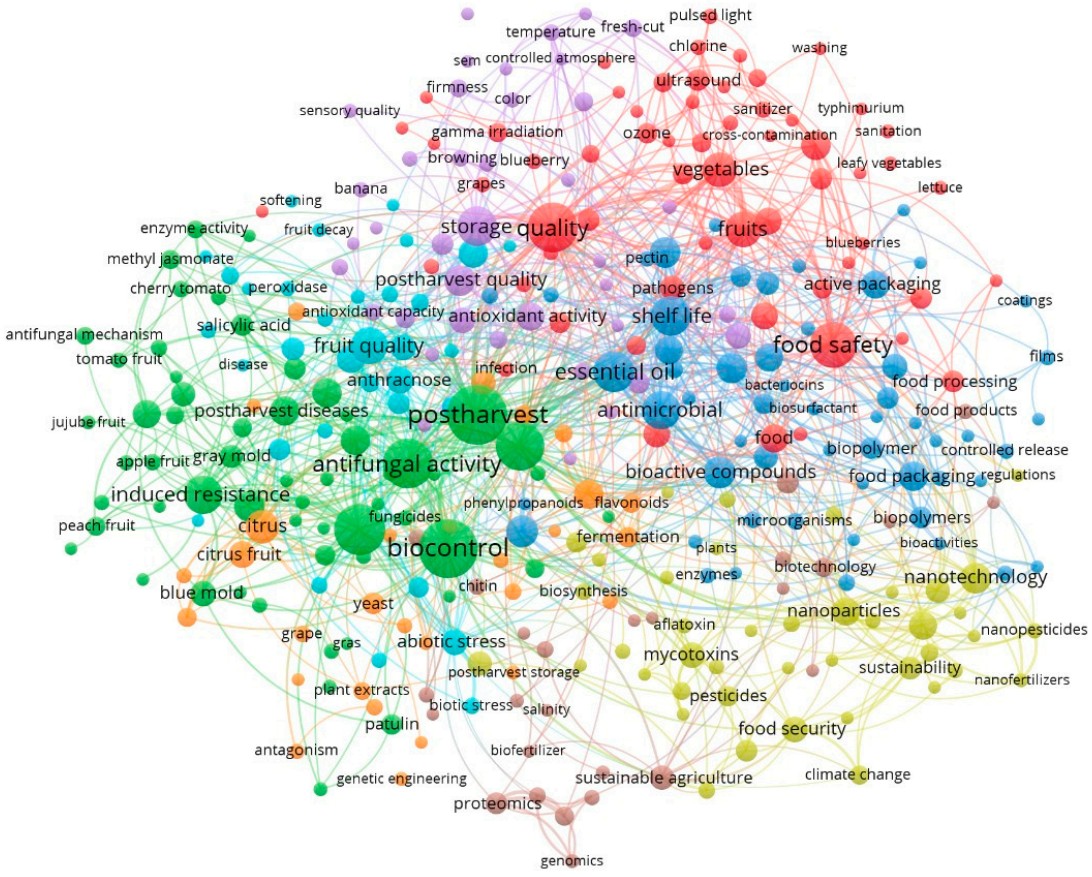

**Figure 1.** Network visualization of keyword co-occurrence analysis.

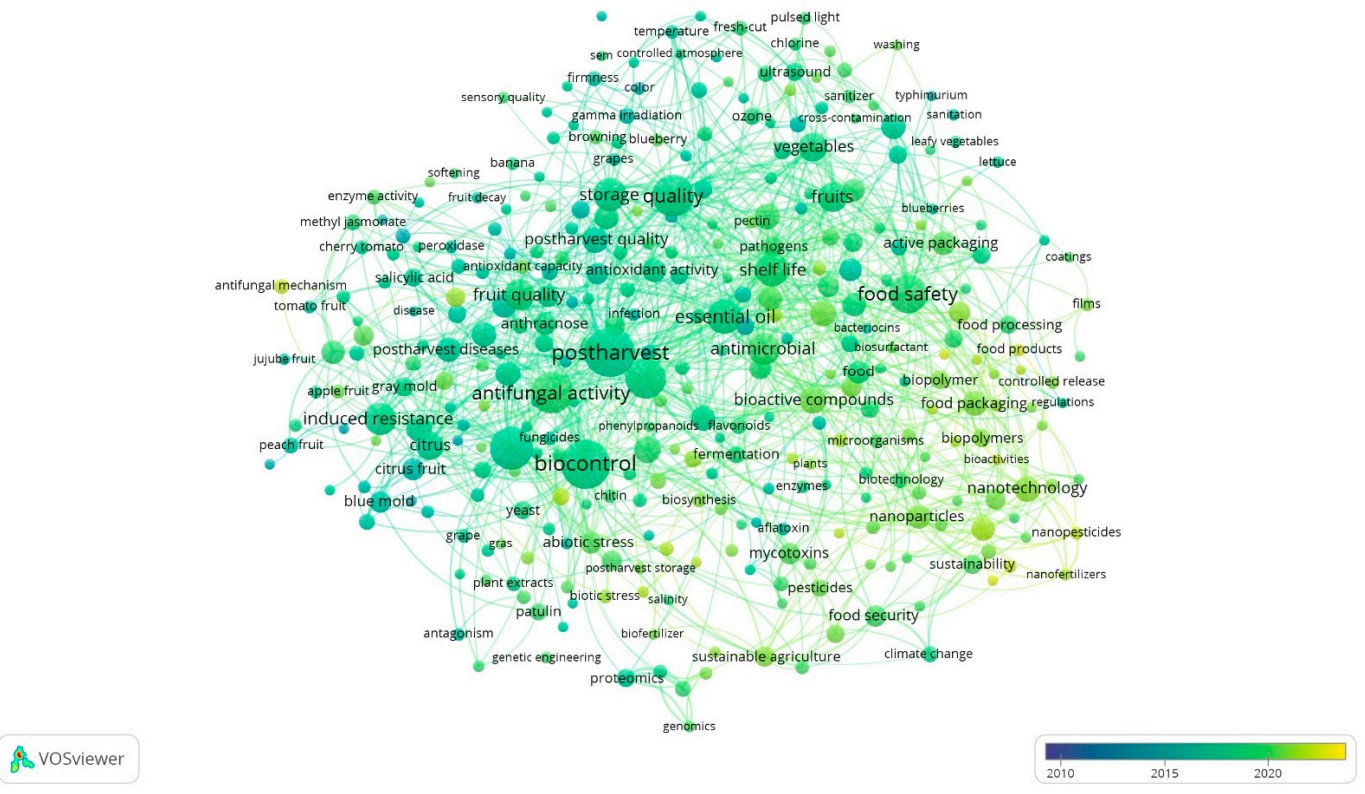

**Figure 2.** Overlay visualization of keyword co-occurrence analysis: time map to investigations 2010–2023.

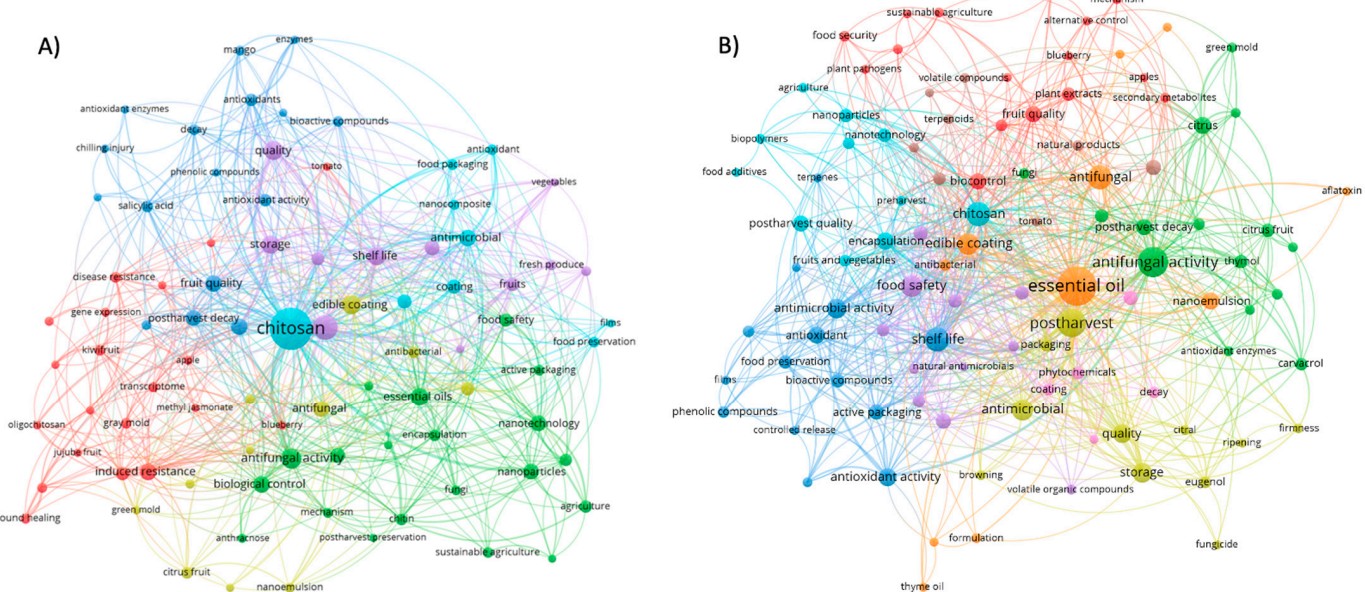

**Figure 3.** Bibliometric analysis of keywords: (**A**) "chitosan", "post-harvest", and "treatments"; and (**B**) "essential oils", "post-harvest", and "treatments". Larger circles refer to topics with greater co-occurrence, and smaller circles refer to topics with less co-occurrence.

According to reports from emerging academic markets, it is estimated that, by the year 2027, alternative treatments could achieve a market size close to USD 28.93 billion. Therefore, the diversity of academic research serves as the spearhead in the quest for new trends and strategies within the industrial sector [28,29].

### 3. Chitosan, the Biopolymer with Multiple Properties

A cationic polymer derived from chitin, chitosan [30], has garnered interest considering its chemical compatibility with other organic reagents and its intriguing mechanisms related to antimicrobial activity and enzyme defense production, particularly regarding post-harvest quality in fruits [31,32].

Obtained primarily from natural sources of chitin such as marine crustacean shells, insects, and certain fungi of the Zygomycetes class [33], chitosan undergoes a thermos-alkaline deacetylation process. This process wields a polymer with a high degree of reactivity, due to the presence of amino and hydroxyl groups within its chemical structure [34–36]. These functional groups, which are the primary hydroxyl group (OH) at position C-6, the secondary hydroxyl group (HO) at position C-3, and the amino group ($NH_2$) at position C-2, contribute significantly to chitosan´s biological properties. Modifications are often made at the C-2 and C-6 positions to enhance its efficacy as a treatment and its ability to bind with other compounds (see Figure 4) [30,37].

**Amino group**
$C_2$ - $NH_2$
- Schiff base modification
- Quaternization
- Alkylation, Acylation, Carboxylation, and Phosphorylation
- Chemical coupling
- Metal coordination
- Graft copolymerization
- Cross-linking

**Primary hydroxyl group**
$C_6$ - OH
- Esterification
- Carboxymethylation
- Phosphorylation
- Alkylation
- O-coupling
- Quaternization
- Graft copolymerization
- Cross-linking
- Metal coordination

**Secondary hydroxyl group**
$C_3$ - HO
- Methylation
- Acylation
- Sulfonation
- Alkanoylation

*B-(1 --> 4)-glycosidic bond*

**Figure 4.** The functional groups of the chitosan molecule and their influence on reactivity.

In addition to its chemical characteristics, several parameters influence the physical properties of chitosan. These include molecular weight, the degree of N-acetylation, solvent evaporation, and the mechanism of regeneration of free amino groups [38]. Furthermore, factors such as the method of application, the food matrix, and the targeted phytopathogen play crucial roles in determining chitosan´s effectiveness [39,40].

*Effect of Chitosan against Phytopathogens in Fruits*

Recently, there have been reports on chitosan's properties during fruit post-harvest storage and its role in promoting plant growth [30]. Consequently, various hypotheses have emerged regarding its mode of action, ranging from its structural effects on pathogenic

cells, resulting in the release of intracellular components [41,42], to its ability to absorb essential nutrients from these cells, leading to a decrease in mRNA and protein synthesis and, ultimately, the inhibition of pathogen development [43,44].

The effectiveness of chitosan as a post-harvest treatment has been attributed to three mechanisms of action: (i) the formation of a physical barrier between plant tissue, pathogens, and environmental microorganisms, affecting gas interactions both internally and externally (see Figure 5A) [33,39,45]; (ii) the reinforcement of plant tissues through the induction of defense enzymes, the stimulation of lignin deposition, and the increased production of antimicrobial phenolic compounds (see Figure 5B) [40,42,46]; and (iii) antimicrobial activity, which exhibits fungistatic or fungicidal effects by permeabilizing and damaging the cell wall and membrane of fungi (see Figure 5C) [30,33,34,36,47]. Moreover, this antimicrobial activity is associated with chelating properties, whereby chitosan captures essential nutrients and metals such as zinc (Zn), copper (Cu), cobalt (Co), manganese (Mn), nickel (Ni), and cadmium (Cd). Once chelated, the positive charges present in chitosan´s amino group are reinforced, enhancing its ability to interact with the surface components of microorganisms (see Figure 5C) [34].

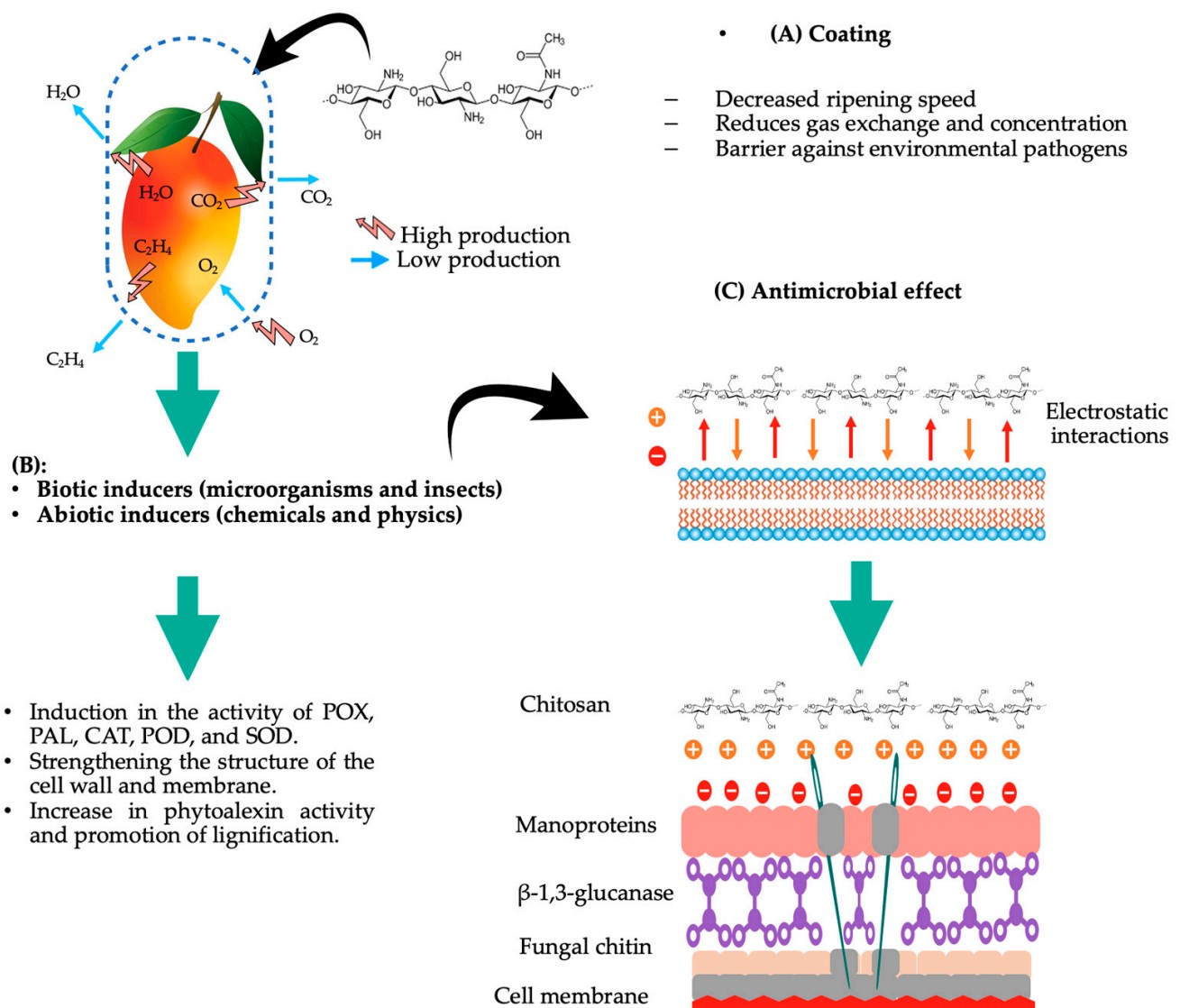

**Figure 5.** Proposed mechanisms of action of chitosan: (**A**) coating effect; (**B**) induction of natural defense mechanisms; and (**C**) antimicrobial effect.

Currently, chitosan is part of a variety of treatments that, through its functionalization, can incorporate various organic and inorganic materials within the polymer matrix [48–51], improving biofilm and antimicrobial properties using commercial products at low concentrations [39,52].

An analysis of the *Kyoto Encyclopedia of Genes and Genomes* [53,54] suggests that chitosan´s potential antimicrobial mechanism involves interactions with the genes responsible for downregulating ribosome formation. Conversely, the genes associated with glycerophospholipid and ether lipid metabolism, as well as steroid biosynthesis, are upregulated. These genes play a role in maintaining cell membrane stability and could hinder fungal development [53–57]. Furthermore, chitosan interacts with a plant´s immune system, triggering a cascade of biochemical reactions and stimulating the activation of genes related to the activity of chitinase, peroxidase, catalase, polyphenol oxidase, and B-1,3-glucanase, while also increasing the concentration of flavonoids and lignin [57–59]. Moreover, the versatility of this polymer with other technologies helps enhance the formation of a rigid network derived from these properties [30,60,61].

## 4. Aromatic Compounds and Their Effects

Recently, the use of eco-friendly technologies in the post-harvest storage of fruits has emerged to reduce reliance on chemical methods, such as synthetic fungicides. Alternative approaches, such as plant extracts, have demonstrated diverse mechanisms for controlling fungal growth by intervening in the host–pathogen interaction [62–64].

Natural extracts can be sourced from various plant parts, including leaves, roots, bark, flowers, or fruits, and are considered compounds of high biological value. These extracts are defined as mixtures obtained from the secondary metabolites of plants, composed primarily of hydrocarbons from polymethylene series, falling under the group of terpenes $(C_5H_8)n$. They also contain aromatic compounds such as alcohols, esters, ethers, aldehydes, and phenolics, which are characterized by their oxygenation [65,66].

These properties have been reported to exhibit antimicrobial effects against pathogenic fungi such as *Fusarium* spp., *Alternaria* spp., *Aspergillus* spp., *Penicillium* spp., and *Rhizopus* spp. [67,68].

### 4.1. Citral's Antifungal Properties and Mechanisms

Citral is a natural isoprenoid, which is composed of two isomers, geranial and neral [69], which are found in essential oil extracts of species such as *Cymbopogon citratus*, *Melissa officinalis*, and *Verbena officinalis*, among others [70].

This compound is considered one of the potential alternative treatments in the agricultural industry due to various mechanisms attributed to its bactericidal, insecticidal, and antifungal activity [71]. Its mechanism of action focuses on inducing changes in the permeability of the cell membrane, causing alterations in the basic functions of microorganisms such as Gram-positive and Gram-negative bacteria, as well as fungal pathogens during the post-harvest stage [72,73].

One of the main effects of citral on the development of phytopathogenic fungi is the dehydration and distortion of hyphae at a superficial level, along with the collapse of spores. Additionally, due to increased cellular permeabilization, citral induces the leakage of molecular substances and lesions in the cellular metabolism, further affecting pathogen development [69,70,74]. The interaction of citral with ergosterol synthesis, the main sterol present in yeasts and fungi, leads to the inhibition of pathogen growth and cell death. This occurs through distillation at C-14 of lanosterol, the biosynthetic precursor of ergosterol, as well as the possible blockage of scalene epoxidase, affecting the permeability of the membrane [75,76] (see Figure 6).

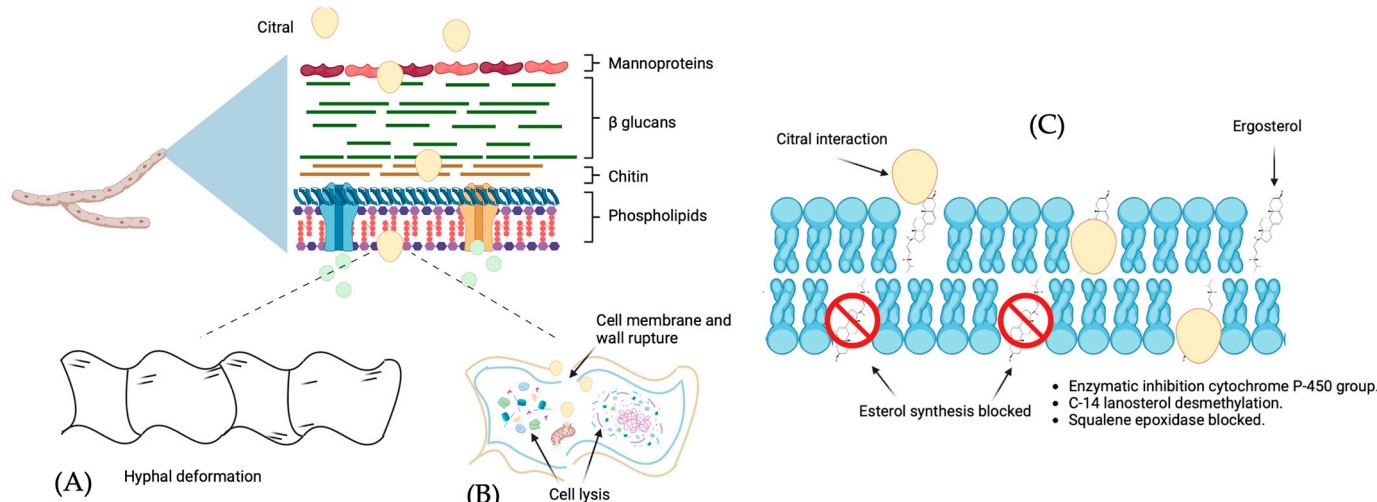

**Figure 6.** Proposed mechanism of action of citral in fungal cells: (**A**) deformation and dehydration of hyphae; (**B**) effects on cellular metabolism; and (**C**) interaction of citral with the ergosterol molecules present in the cell membrane of pathogenic fungi.

Several investigations have tested the efficiency of citral against in vitro pathogens such as *Penicillium* sp. [69], *Geotrichum citri-auranti* [71], *P. italicum* [72], and *Botrytis cinerea* in *Solanum lycopersicum* [70]. Moreover, its efficacy has been evaluated in kiwi fruits [77], tomatoes [78], and citrus fruits [79] (see Table 2). In addition, reports indicate that combining citral with chitosan generates a synergistic effect, enhancing both mechanisms of action against various pathogens [70].

**Table 2.** Citral and its principal mechanisms of action in in vitro applications and in vivo assays.

| Assay | Concentration | Effects | References |
|-------|---------------|---------|------------|
| *In vitro* | 0–200 μg/mL | *Magnaporthe grisea* hyphae exposure to 50 ug/mL showed ultrastructural changes in the morphology, and the application of high concentrations led to severe cellular degeneration; the cell walls appeared to be degraded and displayed cellular disorganization. This proposes that citral ruptures the cell wall and penetrates the cell membrane, as has been seen through scanning and transmission electron microscopy. | [80] |
| *In vitro* | 0.50–1.00 μL/mL | The permeability of the membrane increased in correlation with the concentration of citral; in addition, the application induced a decrease in the content of lipids and ergosterol in *Penicillium italicum* fungal cells. | [69] |
| *In vitro* | 2.0–4.0 μL/mL | Citral application reduces enzymes' activity of citrate synthase, isocitrate dehydrogenase, α-ketoglutarate dehydrogenase, and succionadrogenase; these decreases in mitochondrial enzymes mark a deficiency in the electron transport chain components, a decrease in ATP synthesis and the ability to generate NADPH, and an increase in the oxidative stress in the growth of *Penicillium digitatum*. | [81] |
| *In vitro* | 0–100 μg/mL | Antifungal activity is related to the genes entailed in the chitin and uridine diphosphate synthesis pathways in the amino sugar and nucleotide metabolic pathways of *Magnaporthe oryzae*, causing a reduction in glucan in the cell wall. | [82] |

**Table 2.** *Cont.*

| Assay | Concentration | Effects | References |
|-------|---------------|---------|------------|
| *In vivo* | 0.6 μL/mL | The effect of citral treatment in kiwi fruits causes the antioxidant enzyme system, which includes catalase, peroxidase, and superoxide dismutase, to become active, besides physicochemical parameters which decrease weight loss, softening, and fruit respiration. On the other hand, post-harvest quality is maintained by preventing the breakdown of ascorbic acid content, total flavonoid content, and total phenolic content. | [83] |
| *In vitro* | 0–80 mg/mL | The properties of citral (α-β-unsaturated aldehyde) in the carbonyl group allow β-carbon to become positively polarized and easily reactant to nucleophiles, basing its ability to act as an alkylating agent, capable of influencing biological functions and possibly being harmful by covalently binding to nucleophilic groups within cells. | [73] |
| *In vitro* | 128–256 μg/mL | Citral treatment showed an affinity for ergosterol, inhibited ergosterol biosynthesis, and was related to cell wall alterations, interfering in the cellular metabolism and the loss of membrane integrity, indicating a strong antifungal activity in *Cladosporium sphaerospermum*. | [76] |
| *In vivo* | 0–200 μL/mL | Citral applications in citrus fruits increase the activities of phenylalanine ammonia-lyase, peroxidase, and polyphenol oxidase; moreover, metabolomic analyses induce the accumulation of plant hormones as methyl jasmonate, abscisic acid, and phenylpropanoid metabolites. On the other hand, RNA-seq revealed the expression of multiple genes related to jasmonic acid profiles and phenylpropanoid biosynthesis. | [84] |

Despite its increasing interest as a treatment, its use is limited due to chemical instability, high hydrophobicity, volatility, and susceptibility to oxidative degradation, which may alter its functions [73].

### 4.2. Hexanal's Antifungal Properties and Mechanisms

Hexanal, an organic compound extracted from plants, is produced during lipoperoxidation mediated by lipoxygenase and hydroperoxide lyases. It has been linked to the role of stress-related responses in plants, primarily caused by cellular wounds, leading to the production of toxic agents against various pathogens [85,86]. Belonging to the GRAS group of products, hexanal has been tested in both the pre- and post-harvest stages of fruits [87] (see Table 3). Currently, it is associated with inhibiting the activity of phospholipase D (PLD) [88], an enzyme linked to fruit ripening processes. Its effects include alterations in respiration, color, and firmness, as well as interference with ethylene synthesis. Additionally, hexanal induces mechanisms against pathogenic microorganisms such as *Botrytis cinerea*, *Monilina fructicola* in berries and tomatoes [89], and *Colletotrichum gloeosporioides* and *Lasiodiplodia theobromae* in bananas [90–95].

Considered a natural metabolizable fungicide, hexanal intervenes in lipoxygenase activity, resulting in toxicity for pathogen colonization, reduction in fungal activity, and prevention of new developments, depending on factors such as concentration and exposure time [85,96].

Li et al. [96] demonstrated hexanal's antifungal mechanism against various microorganisms. Their metabolic analysis revealed effects on the tricarboxylic acid (TCA) cycle due to reactive oxygen species (ROS) accumulation, alterations in fatty acids and lipid membranes biosynthesis and composition, mitochondrial activity, increased oxidative stress, and intracellular compound leakage (see Figure 7).

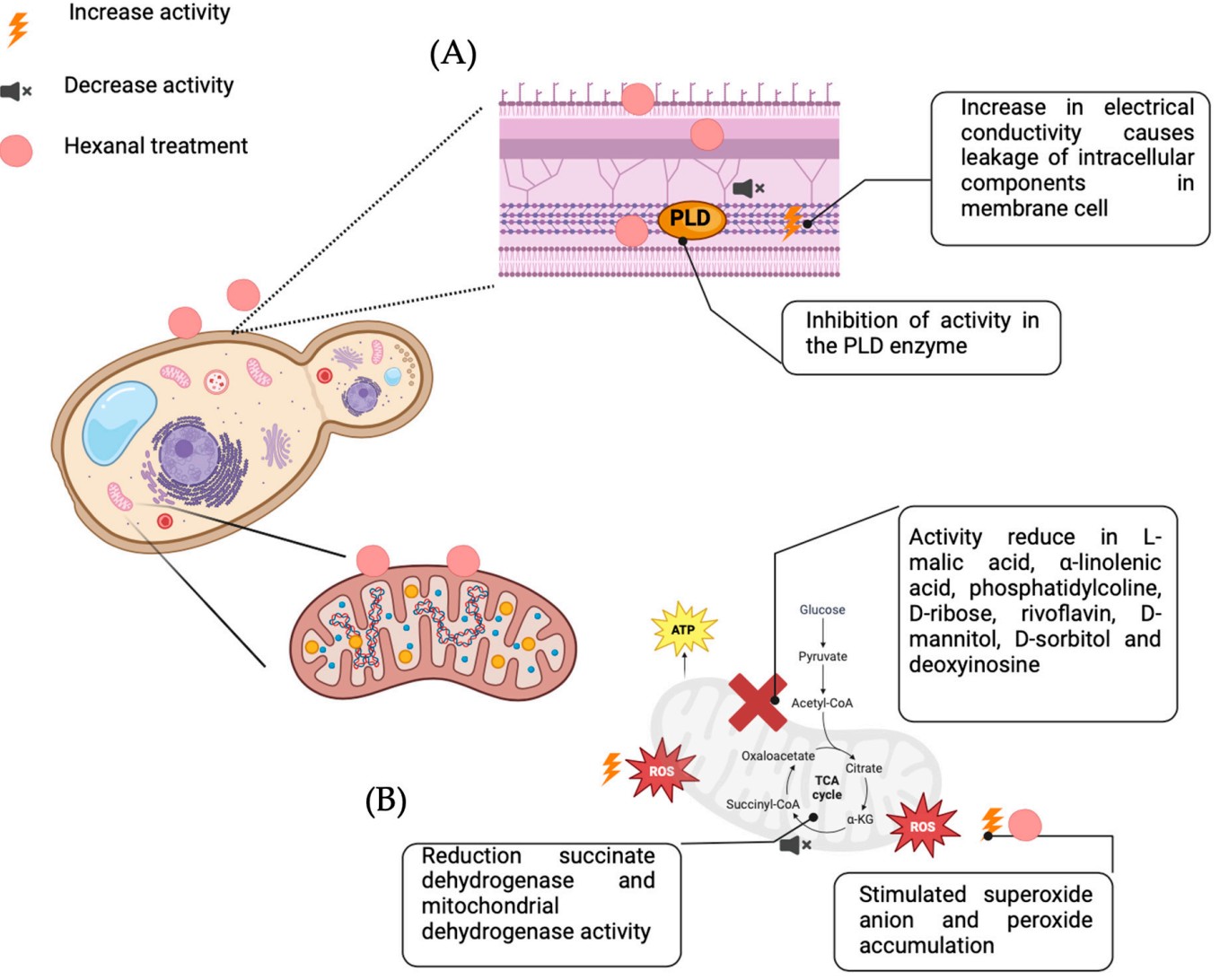

**Figure 7.** Proposed mechanism of action of hexanal against pathogenic microorganism: (**A**) effect on membrane cell and lipophilic enzymes' activity; and (**B**) changes in mitochondrial activity and the TCA cycle.

These activities are integral to fruit quality deterioration during the post-harvest stage, involving elevated ethylene levels. PLD and other lipophilic enzymes are responsible for lipid membrane hydrolysis [86].

**Table 3.** Hexanal's applications pre and post harvest and its principal mechanisms.

| Application | Concentration | Effects | References |
|---|---|---|---|
| Pre- and post-harvest | 2–3% | Pre-harvest spray application in banana var. Grand Nain improves fruit retention by 12–18 days compared with the control. Post-harvest application decreases peroxidase activity and protein synthesis in the abscission zone, delayed the climacteric peak, and decreased the activity of the enzymes that convert stored carbs to soluble sugars. | [97] |

**Table 3.** *Cont.*

| Application | Concentration | Effects | References |
|---|---|---|---|
| Pre-harvest | 800–2000 µM | Hexanal applications to 1600 µM 30 and 15 days before harvesting reduce significantly the incidence of pathogens, pectin methyl-esterase activity, and the respiration rate and delay the activity of phospholipase-D of mango fruits, exhibiting an increase in firmness, total soluble solids, and acidity and acceptable palatability during 28 days at 12 °C in storage. | [98] |
| Post-harvest | 600–1200 ppm | Vapor application at 800 ppm reduces by 75–80% pathogen incidence; on the other hand, it increases peroxidase, polyphenol oxidase, phenylalanine ammonia-lyase, and glucanase activity and phospholipase-D inhibition of the main enzymes in the hydrolysis of phospholipids, thereby increasing the shelf life of fruits and contributing to the phenylpropanoid pathway´s induction of resistance in banana fruits against *Colletotrichum gloeosporioides* and *Lasiodiplodia theobromae*. | [92] |
| Post-harvest | 2.24–2.52 mg/mL | Hexanal concentration showed an inhibitory effect on the growth of *Escherichia coli*. The antimicrobial activity blocking the activity of superoxide dismutase and phospholipase-D inhibition and, combined with heat shock, provoked the overexpression of genes related to fimbria, curli, and biofilm regulation, suggesting that bacteria are induced to stress and are unable to induce biofilm formation in these conditions. | [99] |
| Pre-harvest | 0.02% | An application in apples evidenced that fruit retention and firmness improved, without showing an effect on parameters such as sugar contents and weight. Enzymes that break down the cell wall were less active after being sprayed with hexanal, such as polygalacturonase, glucanase, and gene expression, such expansins. In addition, the authors suggest that hexanal is involved in ethylene biosynthesis, decreasing the expression of four genes related to commercial maturity. | [94] |
| Post-harvest | 0.15, 0.20 y 0.25% | Suppression of cell wall degrading enzymes activity and maintenance of parameters such as firmness, total soluble solids' content, carotenoids, and antioxidant activity on jujube fruits "Umran". The antioxidant enzymes activity, such superoxide dismutase and peroxidase, led to a positively active increase in the commercial life of the fruits up to 21 days in cold storage. | [100] |

## 5. The Ability of Aldehydes and Chitosan to Improve Their Post-Harvest Mechanisms

Considering that the polymeric matrix may serve as a carrier for a wide range of additives, the use of edible coatings for disease management has grown in favor in recent years, and the application of combined treatments has significantly reduced the presence of phytopathogenic microorganisms in many studies [55,101–103].

One of the reactions that has been extensively studied is the Schiff base interaction, which involves the union of two organic compounds derived from a primary amine and an aldehyde or ketone. These interactions result in a double bond at C=N and also act as Lewis bases, expanding their applications in various environments [103]. This reaction facilitates the insertion of the functional groups of the chitosan used as a polymer matrix, enhancing its

properties as an antimicrobial treatment when combined with other treatments possessing specific structural characteristics, such as citral and hexanal [104].

Several investigations have tested treatments made from biphasic elements, which, when mixed or combined, enhance their antimicrobial properties, adhesiveness, biodegradability, and even bioactive potential. These treatments induce physiological changes in fruit to prolong their shelf life during the post-harvest stage [39,105,106].

The pH of the aqueous medium and the aldehyde chemical structure are two essential elements for a stable interaction between chitosan and aldehydes. Using the micro-atmosphere method, combined antifungal effectiveness can be assessed, such as with citral and cinnamaldehyde, which improve polymer cross-linking and prevent *Penicillium expansum* and *Botrytis cinerea* from growing [55,107,108].

Additionally, it has been reported that the polymer matrix, in conjunction with an aldehyde like citral, exhibits a broad spectrum of antifungal activity due to its potential to form charge-transfer complexes with electron donors and its interaction with the SH-groups [109].

## 6. Chitin Derivatives and Commercial Natural Compounds: Their Integration into Post-Harvest Management and Food Sovereignty

In recent years, the focus of technology and research in the agri-food sector has shifted towards products that allow an increase in crop yields, as well as improvements in quality during the pre- and post-harvest stages, utilizing natural, biodegradable, and environmentally benign active substances which do not negatively impact the environment, as established by the United Nations General Assembly in its 2030 Agenda [5,110].

An alternative method that has shown very promising results in its applications, enhancing the development and quality of food matrices by increasing the activity of certain compounds and their response to stimuli such as biotic and abiotic stress, involves products extracted from natural materials [111].

In the commercial sector, biopolymers such as chitosan have gained traction in agriculture, primarily functioning as bio-stimulants in plant development and protection, as well as edible biofilm for fruits during their post-harvest stage, due to their physicochemical and biological properties [60,112].

In the alternative-technologies market, companies such as Lallemand Oenology®, Cultivers eco®, Summit Agro®, and Biorend® are the main suppliers of chitosan as a treatment for improving the development of plants/fruits and controlling diseases caused by phytopathogens. Chitosan is offered at concentrations of 2.5 and 3 g/L and is available in presentations ready for dilution in water and subsequent application [113].

On the other hand, aldehydes such as citral and hexanal are widely used in products for perfumery and cosmetics and as flavorings for some foods. In addition, they play an important role as masking agents in organic synthesis preparations such as vitamin A, ionone, and p-cymene [114,115]. In some experimental stages, they have demonstrated effects on phytopathogen control, increasing natural defense mechanisms as well as improving the physicochemical characteristics of some fruits during their post-harvest stage. However, there is still no commercial product whose active ingredient is citral or hexanal [83,116,117].

It is noteworthy that agroecology, which has food sovereignty and security as its primary pillars, is one of the primary societal commitments focused on producing food free of pests, diseases, and pesticides [118]. Its primary objective is to modify the agri-food system to contribute to the well-being of the population [119], taking into account concepts such as the right of each individual to participate in food production and preservation, as well as the cultural and productive diversity of each community, using food security as the main mean [120].

For this reason, the application of this type of technology, especially in post-harvest stages, can pave the way towards the revaluation of knowledge and techniques that can be innovated to promote multidisciplinary growth. This includes the promotion of programs and projects that are relevant to sustainable production, addressing agri-food problems and

encompassing the set of sectors associated with food production, distribution, marketing, and consumption [121,122].

## 7. "Omics" Sciences and Their Participation in the Knowledge of New Defense Mechanisms and Their Interactions

In natural and storage environments, fruits are exposed to interactions with pathogens with the potential for infection, and various symptoms depend mainly on the host [123]. Infection mechanisms by phytopathogens such as fungi can occur in any part of the plant and are most evident in the post-harvest stage of the fruit. However, the presence of these microorganisms causes damage that is generally associated with quality, such as stains and rot, leading to economic loss and potential hazards if consumed [124–126].

In this context, a complex series of mechanisms is initiated once this interaction between fruit and pathogen occurs. Therefore, recent studies have offered essential information to understand and implement new strategies in disease control during both pre- and post-harvest stages and have gained recognition. Tools such as omics sciences can contribute significantly to the exploration of new knowledge regarding both compatible and non-compatible interactions [127–129].

Recent research on responses in the pathogen–host interaction has been expanding, leading to advancements in the development of products and treatments. This progress is based on various approaches, including diagnostics, disease control products, and the generation of genetically modified organisms [130–132]. However, the most significant advancements are associated with omics sciences. Through genomic, transcriptomic, proteomic, and metabolomic studies [133,134], coupled with activities involving physiological, biochemical, and molecular mechanisms, these sciences allow for the expression and interpretation of genes, as well as the regulation, translation, and metabolic pathways present in plant crops [135].

The applications and new horizons identified with the implementation of omics technologies have improvements in the quality of fruits as one of their main perspectives in the immediate future. Parameters such as firmness, senescence, lignin accumulation, and the activity of secondary metabolites such as phenolic compounds have also demonstrated effects against the development of pathogens [136], allowing us to understand and identify cellular responses to various products, pests, and diseases. These technologies serve as a guide to the possible scenarios occurring in the biological system being evaluated [134,137].

"Omics" studies integrate information about organic molecules, cells, tissues, and metabolic pathways that respond to various exogenous and endogenous factors. Therefore, other disciplines are linked to different applications of biological sciences, such as epigenomics, lipidomics, nutrigenomics, pharmacogenomics, and toxicogenomics. These disciplines collaborate in the search for more specific answers, aiming to enhance the understanding of specific mechanisms of action of the fruits and future treatments [131].

Chitosan is one of the most studied biopolymers in the control of diseases and post-harvest damage in fruits [138,139]. Due to its different mechanisms of action, in transcriptomic analyses and metabolomics, chitosan applications significantly increase the expression of genes and metabolic pathways that interfere with defense systems. As a result, cellular and metabolic processes, as well as the response to biotic and abiotic stimuli, biogenesis, and translation signals, generate effects on the development of phytopathogens in fruits [132,140].

Metabolomic analyses have demonstrated the effect of hexanal application on metabolites related to the TCA cycle and ABC transport systems. This affects membrane synthesis in the pathogenic cells of filamentous fungi and causes an increase in oxidative stress in the mycelium of various fungal species [96,141,142].

## 8. Perspectives and Conclusions

To prevent diseases brought by pathogens in fruits and vegetables, this review paper presents three distinct treatments with GRAS substances. These treatments may have

a better effect when combined, as they could significantly enhance their antimicrobial properties while also improving post-harvest quality. Chitosan, being a stable polymeric matrix, can be combined with natural extracts to create a broad-spectrum edible coating with good mechanical resistance, provided that an appropriate formulation is achieved based on the results already reported for each component. In some cases, combinations of these treatments have already been established, utilizing emerging technology methods to reduce the environmental impact.

Furthermore, the impact of unexplored fields for alternative treatments and the advancements in research using omics technologies open the door to new knowledge in pathogen–host interactions and the identification of new mechanisms or pathways involved in the metabolic or genetic response of fruit to prevent damage and diseases.

**Author Contributions:** Conceptualization E.R.-D. and P.G.-M.; investigation, writing, and draft preparation, E.R.-D.; bibliometric analyses, E.R.-D. and J.A.S.-B.; writing—review and editing, L.G.H.-M., J.A.S.-B., V.M.Z.-G., R.R.G.-E. and P.G.-M. All authors have read and agreed to the published version of the manuscript.

**Funding:** This research received no external funding.

**Data Availability Statement:** Not applicable.

**Acknowledgments:** This work was supported by the National Council of Humanities, Sciences and Technologies (CONAHCYT) for the scholarship granted to Edson Rayón-Díaz.

**Conflicts of Interest:** The authors declare no conflicts of interest.

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
