# Peer review of "Natural Compounds and Derivates: Alternative Treatments to Reduce Post-Harvest Losses in Fruits"

_agriengineering, doi:10.3390/agriengineering6020059_

Round 1

Reviewer 1 Report

Comments and Suggestions for Authors

The main purpose of this work was to analyses the impact of alternative treatments, such as chitosan and essential oil extracts, main citral, and hexanal on post-harvest applications using bibliometric analysis.

This topic can be considered relevant in the area, but some changes/corrections are necessary, which are listed below:

It is known that bibliometric analyses are recognized as crucial tools for tracing the development of a scientific area through literature databases, enabling the identification of highly pertinent and contextually relevant studies that may offer solutions to particular research inquiries.

Title: Page 1

The approach to a bibliometric review is not clear from the title. Therefore, I suggest changing to:

“Alternative treatments to reduce post-harvest losses in fruits: a bibliometric review”.

Or

“A bibliometric analysis of alternative treatments to reduce post-harvest losses in fruits”.

Introduction

References cited in parentheses are in superscript format, please change to normal format. The entire text requires this change.

Page 2, Line 87

Before adding this item, a methodology item is necessary and essential. How can we replicate this data if the methodology is not described? Some questions to this item:

1) The bibliometric analysis was carried out on which platform? Scopus? Web of Science? Another one, which one? It is necessary to identify.

2) If possible, it is interesting to present a PRISMA flow diagram, which summarizes all the search, extraction, filtration and selection process involved in this research.

3) Which Boolean operators were used to combine the general search descriptors? "OR", "AND"?

4)      Furthermore, what fields were used for the search, Title, Summary, Keyword?

5)      What was the period of the search, month/year?

6)      This research includes what type of work, articles published in peer-reviewed journals?  Papers from proceedings? Reviews? Meeting abstracts? Letters? Please identify it.

7)      The obtained results were presented in the form of knowledge domain maps (Figures 1 and 2) visualized using software VOSviewer. Please, it is necessary to add this information, as well as the reference about this software used.

Lines 83-88: This paragraph does not refer to the results of this research, or even the methodology as it is proposed. Therefore, if it remains, it must be added to the Introduction item.

Line 89: “According to a bibliographic search…” This work is not a bibliographical research, but rather a bibliometric one. Please change this.

Lines 89-90: “…using the criteria of “alternative treatments”, “post-harvest”, and “pathogens”, a total of 4,230 results… .” The question is how to ensure that this search refers to fruits? Where is the word "fruit" in this search?

Page 3: Figures 1 and 2: These figures are not legible. Furthermore, it is not clear what this result means from the discussion presented in lines 92 to 102. Generally, this type of bibliometric research presents results such as researchers working in this area, main countries, educational institutions, most used keywords, etc. Presenting only clusters of keywords in recent years does not seem to have much relevance.

Page 4: Figure 2, letters c) and d) do not present relevant results.

In my opinion, the bibliometric research used in this work needs to be better explored.

Another question: Do the authors believe that this manuscript contains an analysis of the impact of alternative treatments such as chitosan extracts and essential oils, mainly citral and hexanal on post-harvest applications through bibliometric analysis? Furthermore, this work presents a discussion about its main properties, effects on fruit application and its role in the agri-food sector and emerging technologies?

Please include everything that was cited in the objective of the work in the conclusions and perspectives item (Page 14).

Author Response

Thanks

Reviewer 2 Report

Comments and Suggestions for Authors

Dear Authors

I have no suggestions to make.

Author Response

Thanks.

Reviewer 3 Report

Comments and Suggestions for Authors

This manuscript describes the impact of alternative treatments, including chitosan and essential oil extracts on post-harvest applications in fruits. The topic is interesting and worthy to be published after minor revision.

1.       In the section “3. Chitosan, the biopolymer with multiple properties”, adding a Figure showing the molecular structure of chitosan will be helpful for readers to well understand the biological properties of the related functional groups.

2.       The subtitle “4. Natural compounds and their effects” sounds not so good, because the previous one “Chitosan” is also a natural product.

3.       Chitosan was repetitively described in both sections 3 and 5. To avoid the repetition descriptions, it is recommended that Sections 5 should be changed as Sections Application Methods, by combining with those relating to Omic.

4.       Overall, it is pity that it lacks the descriptions on data of effects of these alternatives on shelf life of fruits.

Author Response

Thanks.

Round 2

Reviewer 1 Report

Comments and Suggestions for Authors

Dear Authors,

I would like to thank you for the responses and changes made as suggested by this accessory. The article was much better after these changes.

Best regards.